# Spatial-temporal patterns and predictors of timing and inadequate antenatal care utilization in Zambia: A Generalized Linear Mixed Model (GLMM) investigation from 1992 to 2018

**Samson Shumba**[1]*, **Isaac Fwemba**[2], **Violet Kaymba**[3]

**1** Department of Epidemiology & Biostatistics, School of Public Health, University of Zambia, Lusaka, Zambia, **2** University of Zambia School of Medicine, Lusaka, Zambia, **3** Tropical Gastroenterology and Nutrition Group, University of Zambia School of Medicine, Lusaka, Zambia

* samsonshumba1@gmail.com

**Data Availability Statement:** The dataset can be accessed at https://www.dhsprogram.com/data.

## Abstract

Antenatal care (ANC) plays a crucial role in preventing and detecting pregnancy risks, facilitating prompt treatment, and disseminating essential information to expectant mothers. This role is particularly vital in developing countries, where a 4.65% rise in maternal mortality rate was observed in 2022, with over 800 maternal and 7,700 perinatal deaths reported. The study aimed at investigating the spatial temporal patterns and associated factors of timing and inadequacy of antenatal care utilization in Zambia, from 1992 to 2018 using a Generalized Linear Mixed Effect Model (GLMM) approach. The study utilized the Zambia Demographic Health Survey (ZDHS) database. The relationship between dependent and independent variables was examined using the Rao-Scott Chi-square test. Predictors of inadequate utilisation of ANC were identified through the multilevel generalised linear model. Spatial effects were modeled using Quantum Geographic Information System (QGIS) version 3.34.1 to develop univariate choropleth maps. A total of 45, 140 (31, 482 women had less than 8 ANC visits and 2, 004 had 8 or more ANC visits) women with a prior childbirth were included in the study. The findings in the study revealed that among women aged 15 to 19 years in 1992, 1996 and 2013/14, the rates of inadequate antenatal care utilization (less than eight ANC visits) was 90.87%, 90.99%, and 99.63%, respectively. Lack of formal education was associated with inadequate ANC from 1992 to 2018, with percentages ranging from 91.12% to 99.64%. They were notable geospatial variations in the distribution of ANC underutilization across provinces with Luapula, Muchinga, Northwestern, Northern and Eastern recording higher proportions. Furthermore, the study showed that higher education (AOR, 0.30; 95% Confidence Interval, CI, 0.14–0.63; p = 0.001), and wealth index (AOR, 0.34; 95% CI, 0.17–0.70; p = 0.003) correlated with reduced odds of inadequate ANC utilization. In conclusion, this study highlights worrisome trends in ANC utilization in Zambia, with a significant rise in inadequacy, especially in adhering to the WHO's recommended eight ANC visits. Over the period from 1996 to 2018, there was a discernible decline in the

**Funding:** The authors received no specific funding for this work.

**Competing interests:** The authors have declared that no competing interests exist.

prevalence of delayed initiation of ANC. The findings underscore a notable disparity between current ANC practices and established guidelines. Additionally, various factors predicting suboptimal ANC attendance have been identified. These insights call for targeted interventions to address the identified challenges and improve the overall quality and accessibility of ANC services in Zambia.

## Background

Antenatal care (ANC), encompassing the medical support given to a woman throughout her pregnancy, is a fundamental element of reproductive health services. ANC provides a critical opportunity for implementing effective health interventions designed to lower the rates of preventable maternal and new-born mortality (UN, 2014; USAID, 2015) [1, 2]. Each year, about 140 million births occur globally, the proportion attended to by skilled health personnel has increased from 58% in 1990 to 81% in 2019 [3]. Despite a 35% decrease in maternal deaths from 2000 to 2017, dropping from 451,000 to 295,000 annually, the grim reality persists: over 800 women continue to lose their lives daily due to pregnancy or childbirth complications. The majority of these tragic occurrences unfold in sub-Saharan Africa, yet they are largely preventable with access to superior maternal health services [4, 5].

In 2022, Zambia experienced over 800 maternal deaths and more than 7,700 perinatal deaths. The leading causes of maternal deaths include obstetric hemorrhage, hypertensive disorders, pregnancy related infections, abortive outcomes, and other indirect causes and the main cause of perinatal and newborn deaths was asphyxia, prematurity and new-born sepsis remain the leading causes of perinatal and new-born deaths, major causes of [6, 7].The significance of antenatal visits during pregnancy cannot be overstated. Antenatal care (ANC) serves a critical role in preventing and identifying pregnancy risks, ensuring timely treatment, and delivering pertinent information to clients. This is particularly crucial in the context of heightened maternal mortality rates. ANC aims to empower women, fostering a positive childbirth experience [8, 9]. It is essential for protecting the health of women and their unborn children, through preventive health care [6].

The World Health Organization (WHO) in 2016 advocated for a minimum of eight antenatal visits; one contact in the first trimester, two contacts in the second trimester and five contacts in the third trimester [10]. Beyond the recommended frequency, ANC offers a spectrum of benefits, including health checks, risk detection, counselling, and an increased likelihood of skilled birth attendance, thereby playing a pivotal role in reducing both maternal and infant fatalities. As a comprehensive platform for essential services, ANC goes beyond routine checkups to prevent complications, provide crucial birth counselling, and improve overall child health outcomes [11, 12]. Furthermore, the coverage of timely antenatal care (ANC) initiation is low worldwide, with an estimated rate of early ANC visits at 48.1% in low-income countries, compared to 84.8% in high-income countries [13]. In SSA, the initiation of ANC visits within the first trimester is 38% which ranges from 14.5% in Mozambique to 68.6% in Liberia [14]. In Ethiopia, only 28% of women had their first ANC visit during the first trimester and varies across geographic regions [15].

Studies have identified a range of socio-economic and demographic factors influencing the utilization of maternal healthcare services. These factors encompass maternal age, unintended pregnancies, education, wealth status, geographic regions, and residence (urban and rural) [16–22]. In addition, the scarcity of skilled service providers and substandard quality of care

pose significant barriers to maternal healthcare utilization in developing countries. Research indicates that older women are more likely to access healthcare services, possibly due to their experience with health services, confidence in household decision-making, or awareness raised by healthcare professionals about the risks associated with older age [16]. Moreover, low levels of education and a lack of empowerment contribute to the reluctance of women to seek maternal care [17]. The aim of this study was to investigate the spatial temporal patterns and associated factors of timing and inadequacy of antenatal care utilization in Zambia, from 1992 to 2018 using a Generalized Linear Mixed Effect Model (GLMM) approach.

## Methods

This study involved secondary analysis of microdata, utilizing national-level data extracted from the 1992 to 2018 Zambia Demographic and Health Survey (ZDHS) program [23]. The ZDHS is a comprehensive and nationally representative household survey conducted by the Zambia Statistics Agency in collaboration with global partners, including ICF International and the United States Agency for International Development (USAID). Employing a two-stage sampling process, the survey initially selects enumeration areas (EAs) and subsequently households. The nature of the DHS data facilitates the comparison of variables over time, enabling the monitoring of changes in indicators across diverse geographical regions [23].

Participation in the survey was limited to women aged 15–49 years from selected households who had consented to take part in the research. Detailed methods employed in the ZDHS are comprehensively documented [23]. For this specific study, we extracted all pertinent variables from the women's data files (individual recode) 1992–2018 ZDHS dataset. The data under examination pertains to the population of women in reproductive age group with at least one childbirth event. The analysis involved the weighted sample from 1992–2018 DHS. Data was accessed 13th of January, 2024. The authors in this study did not have access to information that could identify individual participants during or after data collection.

### Dependent and independent variables

The focus of this study centered on ANC attendance, categorized into two groups: the primary outcome variable, Inadequate ANC utilization (less than eight ANC visits (coded as "1"), and eight or more ANC visits (coded as "0")). Additionally, a secondary variable, Timing of first ANC ("Delay in first ANC initiation," was examined, with "early" defined as less than four months (coded as "0") and "delayed" as four months or more (coded as "1")). Antenatal care holds paramount significance in averting pregnancy-related complications, providing essential counseling for the well-being of both the mother and the fetus, and ensuring preparedness for a health-facility delivery. Guided by the World Health Organization recommendations from 2018, the study underscored the ideal occurrence of the first ANC visit within the first trimester of gestation, with a minimum of 8 ANC visits throughout the pregnancy. Furthermore, WHO recommended one contact in the first trimester, two contacts in the second trimester and five contacts in the third trimester [10].

Following an extensive literature review, we meticulously controlled for a wide array of demographic, socio-economic, behavioral, and community-level factors in our analysis. Specifically, we included variables that gauged the content of antenatal care, such as discussions on "HIV transmission from mother to child," "preventive measures for HIV," "HIV testing discussions," "receipt of HIV test results during antenatal visits," "counseling post-HIV testing during antenatal care," and "HIV testing as part of antenatal visits," all coded as "yes" or "no." Additionally, the mother's education level was categorized into tiers: 0 for no education, 1 for primary education, 2 for secondary education, and 3 for higher education. Mother's age was

segmented into five-year cohorts, spanning from 15–19 to 45–49. The household's Wealth Index was stratified into five quantiles: 1 for the poorest, 2 for the poor, 3 for the middle, 4 for the rich, and 5 for the richest.

## Data analysis

For descriptive purposes, percentages were computed for categorical variables using sample weighting for accurate representation. To determine association between the outcome variable (primary outcome = inadequate ANC visits "less than eight ANC visits" and secondary outcome = delayed first ANC utilization "first ANC visit greater than 3 months") and the categorical variables, the Uncorrelated Design Based Chi-square test (Rao–Scott Chi-square test) was used. This was selected to consider clustering. To determine the factors associated with inadequacy of ANC visits (having less than eight ANC visits) among women with prior childbirth the study utilized the generalized linear mixed models (GLMMs) with the logit link and binomial family that adjusted for clustering and sampling weights were used to assess the association between the independent variables and the outcome variables (less than eight) [19]. We used a Generalized Linear Mixed Model (GLMM) to account for the hierarchical structure of the data. The variables considered for the random effects were Enumeration Areas (coded as "v001") and ZDHS year (coded as "v000"). Stata version 14.2 was used in the analysis.

## Model equation (GLMM family of binomial)

A Generalized Linear Mixed Model (GLMM) is an extension of the Generalized Linear Model (GLM) that incorporates both fixed effects, which are consistent across all observations, and random effects, which account for variations at different levels of data hierarchy, such as individuals within clusters. In the case where the response variable follows a binomial distribution, the GLMM is particularly suitable for analyzing binary outcomes in complex, correlated, or hierarchical data structures. A link function, typically the logit link, connects the mean of the response variable to the linear predictor. In the context of a binomial family (for binary outcomes), the link function g is typically the logit link function, defined as:

$$g\left(\mu_{ij}\right) = \log\left(\frac{\mu_{ij}}{1 - \mu_{ij}}\right)$$

So, the GLMM equation for a binary outcome with a logit link function can be written as:

$$\log\left(\frac{\mu_{ij}}{1 - \mu_{ij}}\right) = X_{ij}\beta + u_{EA[i]} + v_{Year[i]}$$

Here:

- $\mu_{ij}$ is the probability that the response variable equals 1 for the *ith* observation in the *jth* group.

- $X_{ij}\beta$ represents the fixed effects.

- $u_{EA[i]}$ is the random effect associated with the Enumeration Area for observation *i*.

- $v_{Year[i]}$ is the random effect associated with the year the DHS was collected for observation (Breslow and Clayton, 1993) [24].

**Table 1. Model fitness.**

| Diagnostic Test | |
| --- | --- |
| **Model** | **AIC** |
| Competing model 1 (Null Model) | 13692.81 |
| Competing Model 2 | 2670.41 |
| Accepted Model | 2660.542 |

## Diagnostic test and model fitting

Model 1 (competing model) was the null model without explanatory variables. Model 2 (competing model) included the age of the respondent, education, employment, marital status, wealth index, frequency of listening to the radio, frequency of watching television, and residential area. Model 3 (accepted model) included the DHS year of collection, age of the respondent, education, employment, marital status, wealth index, frequency of listening to the radio, frequency of watching television, residential area, and region. The selection process identified the model with the lowest AIC, signifying optimal fit. Additionally, a likelihood ratio test was conducted to compare the full model (the preferred model) with the nested model (model two). The test yielded a p-value of 0.001, indicating that the full model offers a more accurate and nuanced understanding of the factors influencing ANC inadequacy (as shown in Table 1).

Additionally, the study calculated the intraclass correlation (ICC) to assess the homogeneity of the outcome variable within clusters. The ICC for the level "enumeration areas" is extremely close to zero, indicating that almost none of the variability in the outcome is attributable to differences at this level. In contrast, the ICC for the level "year when the ZDHS was captured given the EAs (year|v001)" was 0.1581604, with a standard error of 0.0457136. This indicates that approximately 15.82% of the total variance in the outcome can be attributed to differences between the "year|v001" groups, holding other factors constant. The 95% confidence interval [0.0874717, 0.2691271] suggests that the ICC is statistically significant, with a relatively precise estimate indicating a notable clustering effect at this level.

To assess multicolinearity among independent factors, the Variance inflation Factor (VIF) was used. There are no concerns with multicolinearity in any of the variables (all VIF<5) as shown in Table 2 below.

## Spatial analysis

To assess the geographical distribution of antenatal care utilization among women aged 15–49 with prior childbirth. The study employed Quantum Geographic Information System (QGIS) version 3.34.1 to generate a univariate choropleth map. The spatial analysis was conducted at the provincial level, aligning each Women of Reproductive Age (WRA) with their respective provincial residence using geo-coordinate data collected during the DHS (1992–2018). This geo-spatial data is based on pre-defined information that assigns each case to a specific province. The unit of spatial analysis was defined as a cluster of sample households, as designated by ZDHS. For consistency, a coordinate system of World Geographic System (WGS) 1984 Universal Transverse Mercator (UTM) Zone 36S was applied to facilitate accurate and standardized spatial representation.

**Table 2. Multicolinearity test.**

| Aug | Sep |
| --- | --- |
| 9902.71 | 14694 |

### Ethics statement

The methodologies employed in 1992–2018 Zambia Demographic and Health Survey (ZDHS), including biomarker measurement protocols, received ethical approval from both the Inner-City Fund (ICF) institutional review boards (IRBs) and the Tropical Diseases Research Centre (TDRC) in Zambia. The consent process includes obtaining informed oral consent from each respondent, and for adolescents under 18 years, consent was obtained from a parent or guardian. Comprehensive information about the DHS consent process is available at https://www.dhsprogram.com/What-We-Do/Protecting-the-Privacy-of-DHS-Survey-Respondents.cfm. Authorization to use the ZDHS data was obtained from ICF Macro and the dataset can be accessed from the Demographic and Health Survey Program (https://www.dhsprogram.com/data). The user meticulously adhered to the provided instructions, with a strong emphasis on maintaining the confidentiality of the data and ensuring the anonymity of the households and individual respondents interviewed in the survey.

## Results

Included in this study were data for 45,140 women, taken from a total sampled population of 59,979. Those without a history of prior child birth (n = 14,839) were excluded. This analysis involved weighted sample from the 1992 to 2018 ZDHS (see Fig 1).

### Characteristics of women who have previously given birth, linked with less than 8 ANC visit

The findings in the study revealed that majority of the women reported having less than eight ANC visits had no level of education throughout the survey years 1992 to 2018 (91.12%, 92.08%, 93.47%, 99.68% and 99.64% respectively). The relationship was significant (p<0.0001). Notably, a significant association (p<0.0001) was found between having less than 8 ANC visits and the absence of formal education, with 91.12%, 92.08%, 93.47%, 99.68%, and 99.64% of women reporting less than eight ANC visits in each respective year.

   Additionally, the study revealed that the majority of participants reporting less than eight ANC visits belonged to the poorest socioeconomic stratum between 2007 and 2013/14 (98.68% and 99.62%, respectively), except for 2018, when a higher proportion (99.08%) was observed among the richer segment. Consistently, women not engaged in employment showed a higher proportion of less than 8 ANC visits from 1992 to 2018, with notable variations in 1996 and 2001/02 as shown in Table 3.

### Characteristics of women who have previously given birth, linked with delayed initiation of their first antenatal care (ANC) visit (four months or more)

The study investigated the characteristics of participants who delayed in the first antenatal care or missed their first trimester ANC visit as shown in Table 4. The study examined factors influencing delayed or missed first ANC visits, with a focus on participant characteristics. The results revealed that a significant percentage of adolescents aged 15 to 19 (80.66%) delayed their first ANC visits compared to older women. This difference was statistically significant (p = 0.0002). Additionally, women with lower education levels (up to primary) exhibited a higher proportion of delayed ANC visits compared to those with higher education (secondary and above). Rural areas had a lower percentage of delayed ANC initiation (76.40%) compared to urban areas (79.86%). Regarding geographic distribution, the central province had the

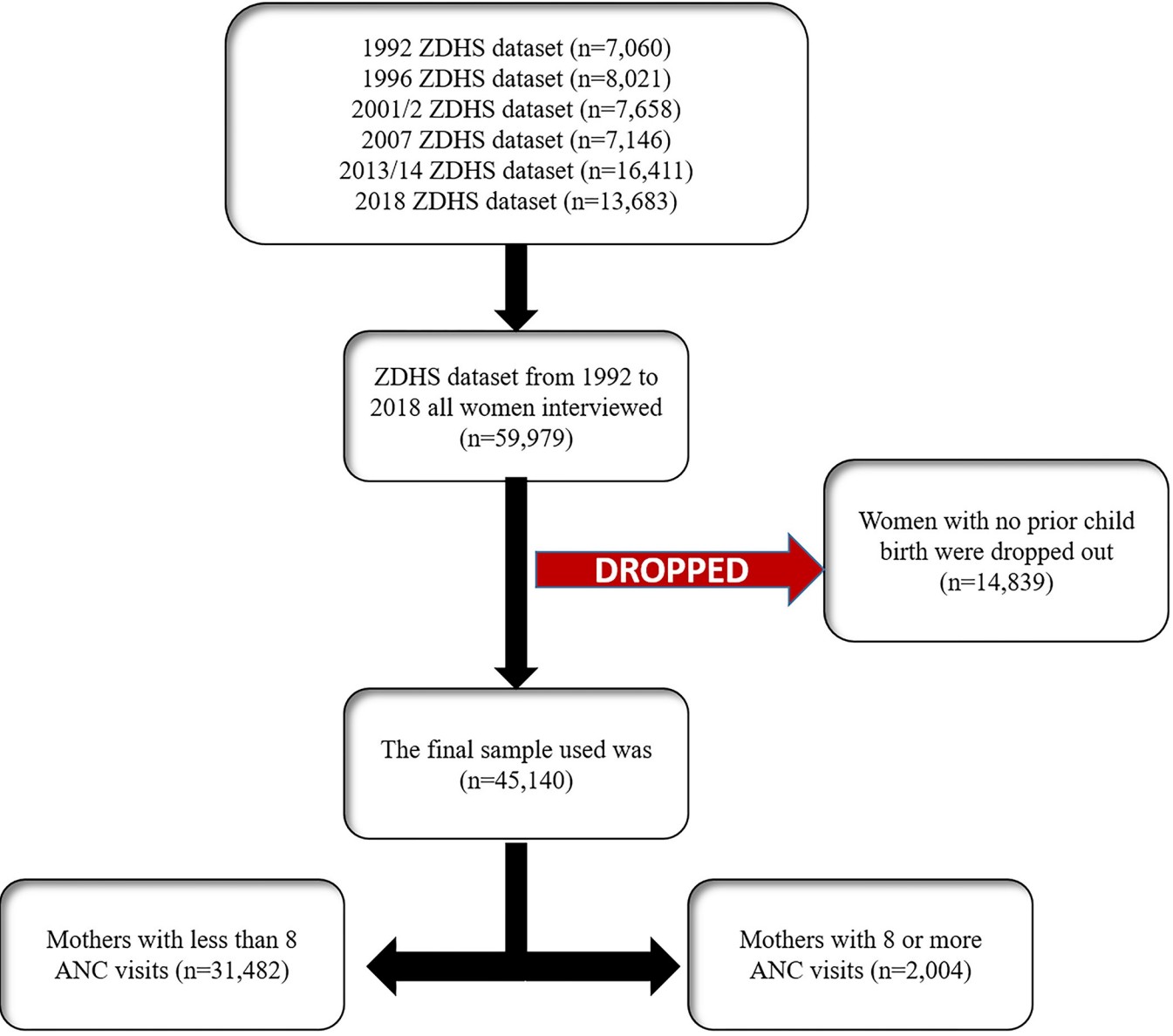

**Fig 1. Description of sample derivation criteria.** For this specific study, we extracted all pertinent variables from the women's data files (individual recode) 1992 to 2018 ZDHS dataset. The data under examination pertains to women with a prior childbirth.

highest proportion of delayed first ANC initiation, followed by Northern (78.72%), Muchinga (78.38%), Copperbelt (78.07%), and Eastern province with the lowest percentage (70.89%), see S1 Table.

### Trends of antenatal care attendance (less than 8 ANC sessions) and delayed ANC initiation from 1992 to 2018

The findings reveal a consistent increase in the proportion of ANC (Antenatal Care) attendance (at least 8 visits) since 1992, with a notable surge observed in 2007. Although there was a slight decline in 2018. Moreover, there was an evident of a consistent decline in the delay of ANC initiation from 1992 to 2018 (see Fig 2).

**Table 3. Key attributes of women who have previously given birth, linked with inadequate ANC visits (less than 8 ANC visit).**

| Variables | 1992 (n = 3,899) | 1996 (n = 4,460) | 2001–02 (n = 4,294) | 2007 (n = 4,031) | 2013/14 (n = 9,234) | 2018 (n = 7,262) | All years (n = 29,333) |
|---|---|---|---|---|---|---|---|
| **Age** | <0.001* | <0.001* | 0.06 | 0.48 | 0.06 | 0.17 | 0.002* |
| 15–19 | 480 (90.87) | 421 (90.99) | 410 (88.53) | 336 (98.83) | 830 (99.63) | 696 (97.98) | 3174 (95.07) |
| 20–24 | 920 (86.04) | 1206 (89.63) | 1086 (88.69) | 98 (97.87) | 2092 (99.19) | 1835 (98.17) | 8137 (94.21) |
| 25–29 | 736 (81.67) | 843 (84.02) | 933 (87.14) | 1064 (97.23) | 2237 (99.22) | 1657 (99.36) | 7469 (93.47) |
| 30–34 | 517 (79.61) | 670 (83.52) | 564 (82.6) | 769 (97.2) | 1858 (98.35) | 1315 (98.91) | 5694 (92.65) |
| 35–39 | 341 (81.1) | 425 (84.39) | 412 (85.76) | 472 (96.82) | 1266 (98.11) | 1037 (98.58) | 3952 (93.5) |
| 40–44 | 207 (84.51) | 199 (81.71) | 243 (87.71) | 250 (96.83) | 664 (99.08) | 487 (98.72) | 2049 (93.7) |
| 45–49 | 76 (89.73) | 82 (83.9) | 82 (86.29) | 91 (99.1) | 186 (98.81) | 144 (34.29) | 662 (93.97) |
| **Education level** | <0.001* | <0.001* | <0.001* | <0.001* | <0.001* | <0.001* | <0.001* |
| No Education | 609 (91.12) | 572 (92.08) | 560 (93.25) | 525 (98.47) | 944 (99.68) | 681 (99.64) | 3891 (95.99) |
| Primary | 2075 (85.6) | 2511 (89.47) | 2376 (89.47) | 2448 (98.04) | 4925 (99.45) | 3501 (98.57) | 17836 (94.43) |
| Secondary | 550 (74.63) | 717 (76.75) | 747 (78.09) | 901 (96.84) | 2922 (98.71) | 2689 (99.08) | 8527 (92.35) |
| Higher | 42 (61.91) | 46 (47.69) | 48 (58.21) | 105 (86.8) | 331 (90.87) | 300 (95.88) | 873 (83.47) |
| **Marital status** | 0.13 | 0.03* | 0.93 | 0.77 | 0.55 | 0.87 | 0.48 |
| Not married | 714 (86.06) | 790 (84.19) | 771 (86.98) | 754 (97.67) | 1929 (99.08) | 1809 (98.67) | 6766 (93.9) |
| Married | 2564 (83.53) | 3056 (86.81) | 2959 (86.84) | 97 (97.44) | 84 (98.85) | 5362 (98.77) | 24370 (93.64) |
| **Wealth Index** | | | | <0.001* | <0.001* | 0.32 | <0.001* |
| Poorest | – | – | – | 903 (98.68) | 2031 (99.62) | 1635 (98.98) | 4569 (99.2) |
| Poorer | – | – | – | 839 (98.41) | 1931 (99.57) | 1497 (99.07) | 4267 (99.17) |
| Middle | – | – | – | 820 (98.66) | 1888 (99.22) | 1361 (98.55) | 4070 (98.88) |
| Richer | – | – | – | 813 (97.06) | 1769 (98.99) | 1448 (99.08) | 4070 (98.63) |
| Richest | – | – | – | 604 (93.6) | 1513 (96.62) | 1230 (97.88) | 3347 (96.51) |
| **Employed** | 0.17 | <0.001* | 0.03* | 0.003* | 0.04* | 0.06 | 0.001* |
| No | 1542 (84.94) | 1937 (87.96) | 1505 (87.96) | 2013 (98.24) | 4160 (99.25) | 3765 (99.08) | 14921 (94.32) |
| Yes | 1735 (83.3) | 1907 (84.58) | 2222 (84.58) | 1960 (96.72) | 4936 (98.6) | 3406 (98.38) | 16166 (93.12) |
| **Frequency of listening to rad** | | | <0.001* | 0.004* | 0.53 | 0.59 | <0.001* |
| Not at all | – | – | 1814 (91.44) | 1363 (98.52) | 3802 (99.15) | 3967 (98.79) | 10947 (97.58) |
| <once a week | – | – | 566 (88.7) | 482 (98.54) | 986 (98.69) | 894 (99.08) | 2928 (96.68) |
| ≥ once a week | – | – | 520 (87.27) | 546 (96.25) | 1559 (98.87) | 1093 (98.82) | 3718 (96.67) |
| Almost everyday | – | – | 827 (77.07) | 1587 (96.72) | 2780 (98.64) | 1215 (98.29) | 6408 (94.69) |
| **Frequency of reading NS/MZ** | | | <0.001* | 0.002* | <0.001* | 0.17 | <0.001* |
| Not at all | – | – | 3177 (89.54) | 2936 (97.94) | 6599 (99.48) | 5982 (98.83) | 18694 (97.2) |
| <once a week | – | – | 337 (77.47) | 442 (97.02) | 1071 (97.02) | 654 (98.99) | 2504 (94.31) |
| ≥ once a week | – | – | 178 (73.68) | 443 (27.88) | 1001 (98.54) | 418 (97.79) | 2039 (95.24) |
| Almost everyday | – | – | 34 (52.80) | 154 (92.15) | 437 (95.71) | 118 (96.49) | 743 (91.72) |
| **Frequency of watching television** | | | <0.001* | <0.001* | <0.001* | 0.27 | <0.001* |
| Not at all | – | – | 2920 (90.68) | 2896 (98.2) | 5806 (99.37) | 4645 (98.93) | 16267 (97.36) |
| <once a week | – | – | 189 (82.7) | 218 (98.74) | 419 (98.8) | 365 (99.53) | 1192 (96.04) |
| ≥ once a week | – | – | 170 (78.21) | 145 (96.27) | 594 (99.15) | 393 (98.18) | 1302 (95.21) |
| Almost everyday | – | – | 448 (71.7) | 720 (94.6) | 2306 (97.68) | 1768 (98.24) | 5240 (94.52) |
| **Type of residence** | <0.001* | <0.001* | <0.001* | <0.001* | <0.001* | 0.018* | <0.001* |
| Urban | 1411 (76.78) | 1416 (77.99) | 1128 (77.35) | 1257 (95.32) | 3424 (98.01) | 2749 (98.20) | 11385 (89.48) |
| Rural | 1867 (90.56) | 2431 (91.99) | 2602 (91.77) | 2723 (98.52) | 5709 (99.44) | 4422 (99.09) | 19754 (96.32) |

NS = Newspaper; MZ = Magazine; Rad = Radio

* = P-value <0.05 computed using the Rao-Scott Chi-square test.

**Table 4. Key attributes of women who have previously given birth, linked with delayed initiation of their first ANC visit (first ANC session ≥ 4).**

| Variables | 1992 (n = 3,718) | 1996 (n = 4,369) | 2001–02 (n = 4,197) | 2007 (n = 4,083) | 2013/14 (n = 9,156) | 2018 (n = 7,233) | All years (n = 28,985) |
|---|---|---|---|---|---|---|---|
| **Age** | 0.21 | 0.01* | – | 0.02* | 0.37 | 0.68 | 0.001* |
| 15–19 | 448 (90.63) | 412 (92.43) | 386 (87.10) | 288 (85.36) | 634 (76.42) | 463 (65.08) | 2631 (80.66) |
| 20–24 | 922 (90.80) | 1188 (90.25) | 1013 (84.85) | 828 (81.95) | 1595 (75.91) | 1157 (61.82) | 6704 (78.78) |
| 25–29 | 786 (89.83) | 874 (88.36) | 913 (85.52) | 825 (76.44) | 1643 (72.97) | 1023 (61.77) | 6064 (76.58) |
| 30–34 | 553 (88.78) | 689 (86.27) | 574 (84.42) | 631 (80.52) | 1393 (75.07) | 848 (63.77) | 4687 (77.22) |
| 35–39 | 347 (86.02) | 426 (86.82) | 378 (83.46) | 392 (81.19) | 966 (75.66) | 640 (61.84) | 3149 (76.03) |
| 40–44 | 205 (88.67) | 206 (85.11) | 230 (85.33) | 208 (81.97) | 503 (76.40) | 318 (65.71) | 1671 (78.08) |
| 45–49 | 70 (93.15) | 75 (85.56) | 75 (83.66) | 67 (78.63) | 142 (78.46) | 93 (64.10) | 521 (78.67) |
| **Education level** | 0.008* | <0.001* | – | <0.001* | <0.001* | <0.001* | <0.001* |
| No Education | 490 (87.92) | 488 (86.43) | 456 (83.17) | 400 (79.22) | 671 (74.20) | 391 (59.18) | 2895 (77.43) |
| Primary | 2122 (90.57) | 2463 (89.93) | 2274 (87.47) | 2026 (82.22) | 3735 (76.07) | 2146 (60.46) | 14767 (79.37) |
| Secondary | 663 (88.85) | 852 (88.61) | 791 (82.27) | 735 (78.40) | 2237 (75.50) | 1840 (67.95) | 7118 (76.72) |
| Higher | 55 (78.66) | 65 (63.07) | 48 (54.97) | 77 (61.73) | 227 (61.69) | 166 (52.50) | 637 (59.67) |
| **Marital status** | 0.82 | 0.24 | | 0.61 | 0.51 | 0.001* | 0.082 |
| Not married | 728 (89.82) | 819 (87.38) | 749 (85.91) | 616 (79.58) | 1491 (77.28) | 1220 (66.88) | 5623 (78.67) |
| Married | 2603 (89.55) | 3048 (88.87) | 2820 (84.82) | 2622 (80.50) | 5386 (74.53) | 3322 (61.42) | 19802 (77.49) |
| **Wealth Index** | | | | 0.001* | 0.014* | <0.001* | <0.001* |
| Poorest | – | – | – | 737 (82.92) | 1492 (74.83) | 911 (55.68) | 3140 (69.48) |
| Poorer | – | – | – | 659 (79.35) | 1482 (76.95) | 912 (60.51) | 3054 (71.61) |
| Middle | – | – | – | 666 (81.64) | 1424 (75.08) | 892 (64.75) | 2983 (72.91) |
| Richer | – | – | – | 700 (82.90) | 1364 (77.17) | 1013 (69.70) | 3078 (75.68) |
| Richest | – | – | – | 476 (73.03) | 1114 (70.94) | 813 (64.71) | 2403 (69.08) |
| **Currently working** | 0.57 | 0.42 | – | 0.19 | 0.05 | 0.43 | 0.43 |
| No | 1567 (89.93) | 1934 (89.02) | 1463 (84.56) | 1644 (81.20) | 3191 (76.23) | 2390 (63.33) | 12188 (77.98) |
| Yes | 1763 (89.32) | 1933 (88.09) | 2104 (85.40) | 1589 (79.49) | 3659 (74.16) | 2151 (62.21) | 13199 (77.54) |
| **Frequency of listening to rad** | | | | 0.24 | 0.20 | 0.50 | – |
| Not at all | – | – | 1630 (86.20) | 1104 (82.11) | 2836 (74.91) | 2540 (63.69) | 8111 (73.67) |
| <once a week | – | – | 540 (86.73) | 379 (78.60) | 768 (78.13) | 547 (60.53) | 2233 (74.68) |
| ≥ once a week | – | – | 523 (87.35) | 457 (81.08) | 1192 (75.93) | 687 (62.00) | 2859 (74.44) |
| Almost everyday | – | – | 872 (80.73) | 1298 (79.23) | 2076 (73.84) | 769 (62.29) | 5015 (74.14) |
| **Frequency of reading NS/MZ** | | | | <0.001* | <0.001* | 0.55 | – |
| Not at all | – | – | 2971 (86.45) | 2407 (81.78) | 4946 (75.58) | 3767 (62.53) | 14091 (74.36) |
| <once a week | – | – | 355 (81.05) | 371 (81.11) | 855 (77.23) | 434 (65.80) | 2015 (75.69) |
| ≥ once a week | – | – | 201 (80.12) | 331 (72.12) | 762 (74.58) | 266 (62.26) | 1559 (72.26) |
| Almost everyday | – | – | 36 (55.33) | 125 (74.60) | 296 (64.54) | 75 (61.90) | 532 (65.48) |
| **Frequency of watching television** | – | – | – | 0.003* | 0.11 | 0.17 | – |
| Not at all | – | – | 2690 (86.42) | 2352 (81.30) | 4362 (75.49) | 2884 (61.66) | 12288 (74.65) |
| <once a week | – | – | 196 (86.41) | 186 (85.24) | 338 (79.33) | 232 (62.85) | 951 (76.76) |
| ≥ once a week | – | – | 196 (87.95) | 120 (81.22) | 449 (75.58) | 260 (64.98) | 1018 (75.02) |
| Almost everyday | – | – | 189 (76.94) | 579 (75.07) | 1721 (73.24) | 1166 (65.28) | 3957 (71.36) |
| **Type of residence** | 0.025* | 0.039* | <0.001* | 0.13 | 0.26 | <0.001* | <0.001* |
| Urban | 1676 (90.95) | 1669 (89.99) | 1216 (83.23) | 1041 (78.73) | 2645 (76.06) | 1936 (69.46) | 10184 (79.86) |
| Rural | 1655 (88.29) | 2200 (87.50) | 2353 (86.01) | 2197 (81.28) | 4231 (74.53) | 2606 (58.62) | 15243 (76.40) |

NS = Newspaper; MZ = Magazine; Rad = Radio

* = P-value <0.05 computed using the Rao-Scott Chi-square test.

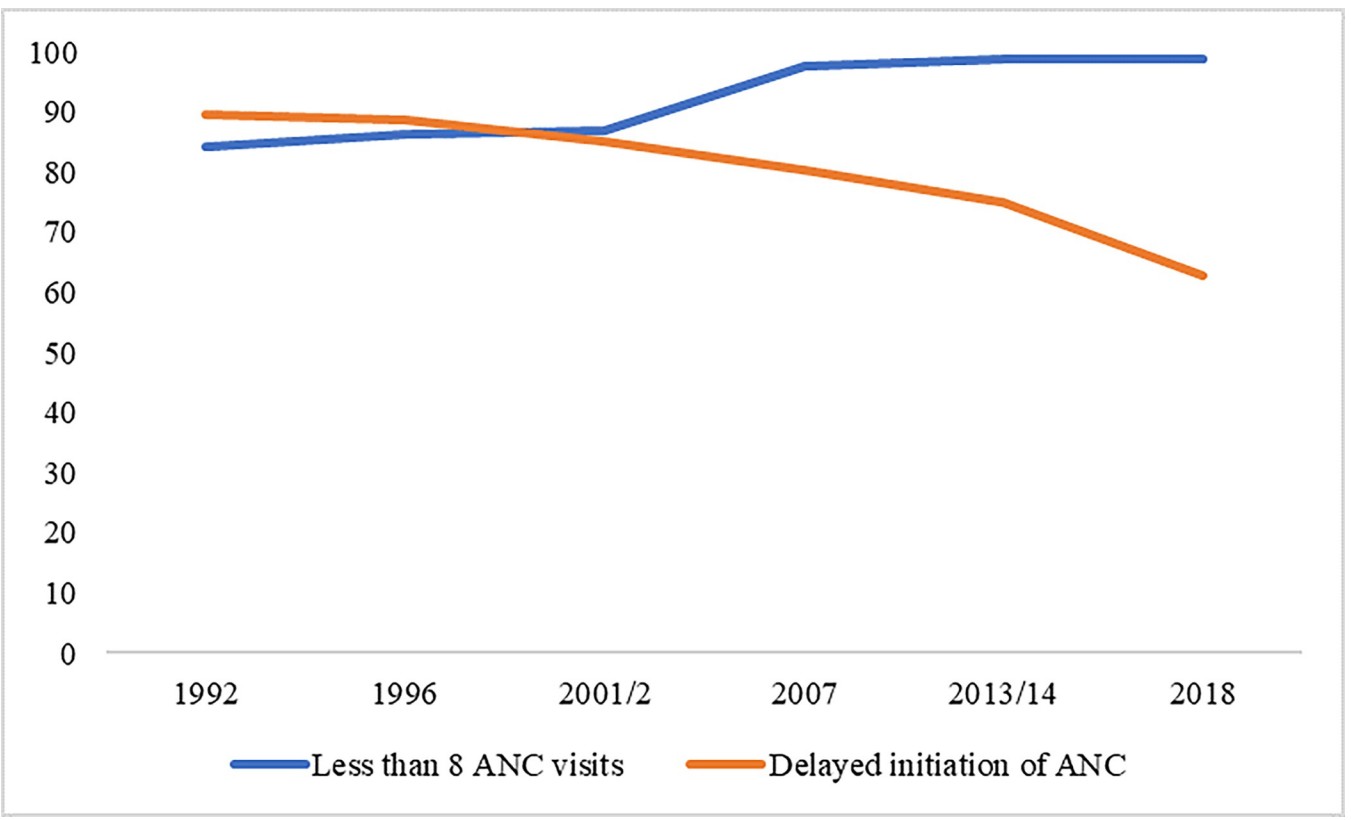

**Fig 2. Trends of antenatal care attendance (less than 8 ANC sessions) and delayed ANC initiation from 1992 to 2018.** The figure was generated in excel. The figure shows the proportion of ANC visits of less than 8 from 1992 to 2018.

### Content of antenatal care offered between 2007 and 2018

The study's findings reveal an upward trend in the quality of services provided as part of antenatal care since 2007. By 2018, all services encompassed within antenatal care were consistently performing at levels exceeding 90%. Particularly noteworthy increase in HIV testing rates, rising from approximately 46% to an impressive 98%. However, the study identified a slight decline in two key variables: discussions about HIV transmission from mother to child and discussions about preventive measures against HIV, showing a decrease of 3.24% and 1.94%, respectively (see Fig 3).

### Spatial-temporal patterns of ANC visits less than eight visits

The study findings reveal notable variations in the distribution of ANC visits across provinces. Generally, Western, Southern, Copperbelt, and Lusaka provinces exhibited a relatively lower proportion of ANC visits less than 8, as compared to other provinces, except in 2018 when Eastern, Western, and Southern provinces outperformed Lusaka. Moreover, regions with higher proportions varied, including Luapula, Muchinga, Northwestern, Northern, and Eastern. Furthermore, Northwestern province remains the only province in the recent years 2013 to 2018 exhibited a higher proportion of the population with less than 8 ANC visits consistently. These variations in ANC utilization proportions were found to be statistically significant using the Chi-square (p<0.001) (see Figs 4–9).

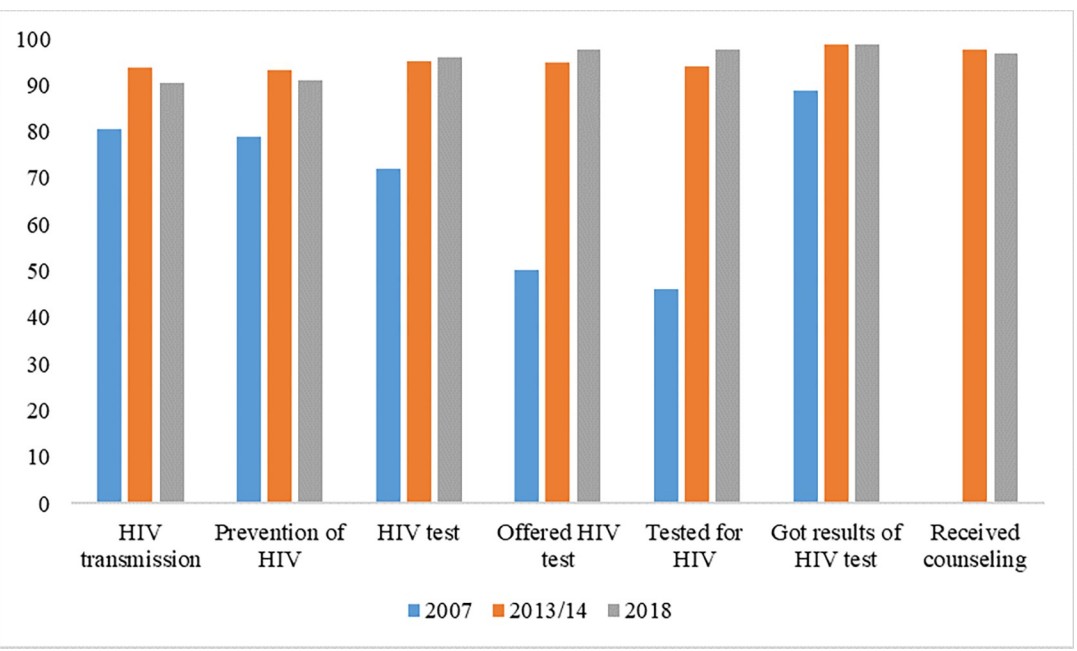

**Fig 3. Content of antenatal care offered between 2007 and 2018.** The variables captured to measure content of antenatal care were (1) talked about HIV transmission of mother to child (2) talked about things to do to prevent getting HIV (3) talked about getting tested for HIV (4) got results of HIV test as part of antenatal visit (5) received counseling after tested for HIV during antenatal care (6) tested for HIV virus as part of antenatal visit.

### Generalized Linear Mixed Model (Combined analysis)

Women with a higher level of education exhibited significantly reduced odds of attending less than 8 ANC visits compared to those with no formal education (Adjusted Odds Ratio, AOR, 0.30; 95% Confidence Interval, CI, 0.14–0.63; p = 0.001), while accounting for all other variables. Similarly, women from wealthier households (richest wealth index) had decreased odds of having less than 8 ANC visits in comparison to those from lower wealth indices (AOR, 0.34; 95% CI, 0.17–0.70; p = 0.003).

Additionally, women engaged in work demonstrated lower chances of having less than 8 ANC visits compared to those who were not working (AOR, 0.66; 95% CI, 0.50–0.87; p = 0.003), and this association was statistically significant when adjusting for other factors. Moreover, women who regularly read news or magazines (almost every day) had reduced odds of attending less than 8 ANC sessions (AOR, 0.59; 95% CI, 0.35–0.97; p = 0.04). However, women who frequently watched television (almost every day) had increased odds of attending less than 8 ANC sessions, holding the effect of other predictors constant. Furthermore, women from Copperbelt province exhibited decreased odds of inadequate ANC compared to women from Central province (AOR, 0.47; 95% CI, 0.27–0.83; P = 0.01) (as shown in Table 5).

### Multivariable Generalized Linear Mixed Model from 1992 to 2018 of less than 8 ANC visit antenatal care utilization (Stratified analysis)

A comprehensive stratified analysis was conducted for each specific year from 1992 to 2018 using data from the Zambia Demographic and Health Surveys (ZDHS). The findings, detailed in Table 5, revealed that older age during selected survey years (1992 and 1996) was associated with reduced odds of less than 8 ANC utilization, and this association was statistically significant (p < 0.05). However, a significant shift occurred in 2018, revealing that older age was

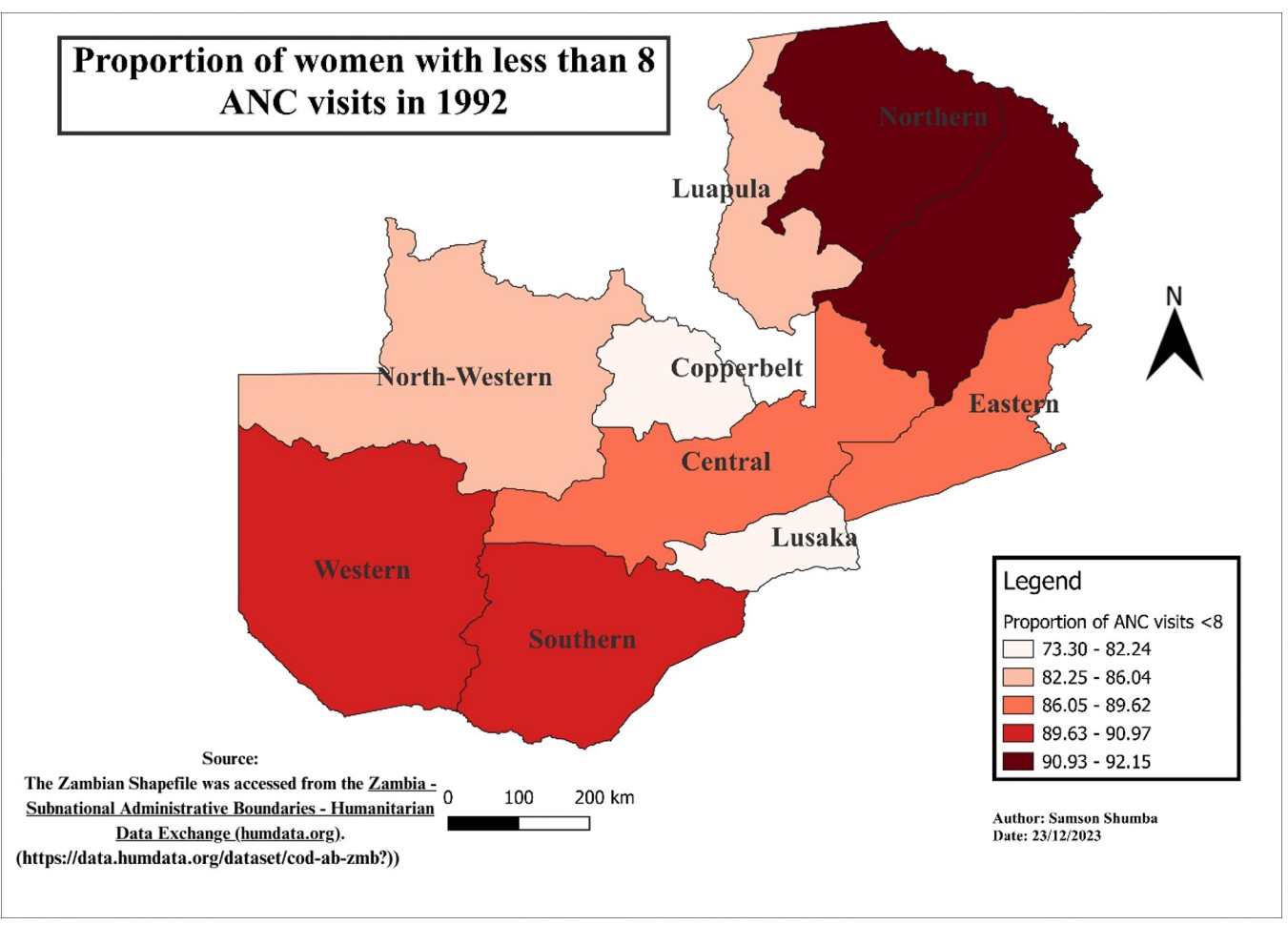

**Fig 4. Shows the spatial-temporal patterns of mothers with ANC visits less than 8 by provinces in Zambia in 1992.** The maps were generated using QGIS version 3.4.1. (Source: author generated).

now linked to an increased risk of less than 8 ANC utilization. Specifically, the adjusted odds ratios for age groups AOR(25–29) and AOR(30–34) were 2.94 (95% CI, 1.20–7.19) and 2.65 (95% CI, 1.05–6.65), respectively. In 2018, older women had increased odds of having less than 8 antenatal care visits compared to women aged 15 to 19.

Furthermore, women with a higher level of education consistently demonstrated a reduced risk of less than 8 ANC visits across the years, except for 2013, where this association was not statistically significant, holding all other factors constant. Moreover, women in rural settings were more at risk of less than 8 ANC visits, and statistically significant predictors were observed only in the period from 1992 to 2001/02. The adjusted odds ratios and corresponding confidence intervals for the respective years were as follows: AOR(1992) was 2.03 (95% CI, 1.44–2.87), AOR(1996) 1.89 (95% CI, 1.43–2.30), and AOR(2001/02) 1.68 (95% CI, 1.21–2.54) (as shown in Table 6).

## Discussion

Antenatal care (ANC) remains a critical area of focus due to its profound impact on maternal and child health, including documented effects on intermediate variables such as birth weight [16]. This study investigated predictors of less than eight ANC visits, and among women aged

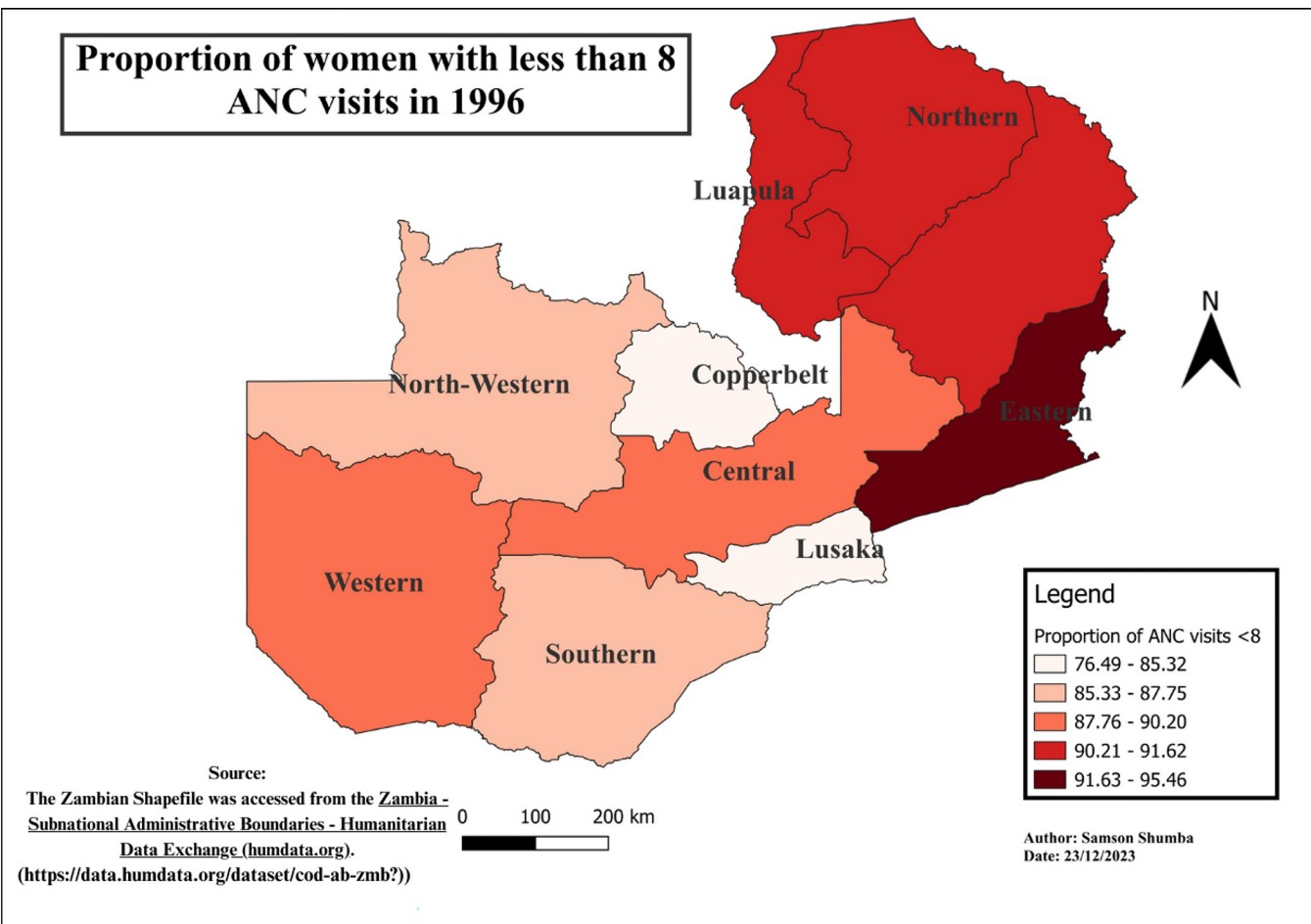

**Fig 5. Shows the spatial-temporal patterns of mothers with ANC visits less than 8 by provinces in Zambia in 1996.** The maps were generated using QGIS version 3.4.1. (Source: author generated).

15 to 49 with prior childbirth in Zambia. Using a multilevel generalized linear mixed model with a binomial family and logit link function on data spanning from 1992 to 2018 from the Zambia Demographic and Health Surveys (ZDHS), the research aims to comprehensively understand predictors and the geospatial distribution of the timing and inadequacy of ANC visits among women aged (15 to 49).

The study's findings revealed an upward trend, an increment of 14% from 1992 to 2018 of women reporting less than eight ANC visits. This indicates a significant gap, consistent between urban and rural settings, reflecting unsatisfactory compliance with WHO-recommended ANC visit levels in Zambia. These findings align with studies in Bangladesh [25], emphasizing the need for improved primary healthcare, especially in rural settings. Furthermore, the study explored the content of ANC, demonstrating an improvement from 2007 to 2018. It stresses the importance of enhancing healthcare content over time, in contrast to some countries, like Latin America, which struggle with low ANC coverage [26]. Therefore, efforts to bridge the urban-rural gap healthcare policies should focus on improving accessibility in rural areas, recognizing the vital role of primary healthcare in promoting ANC in these settings.

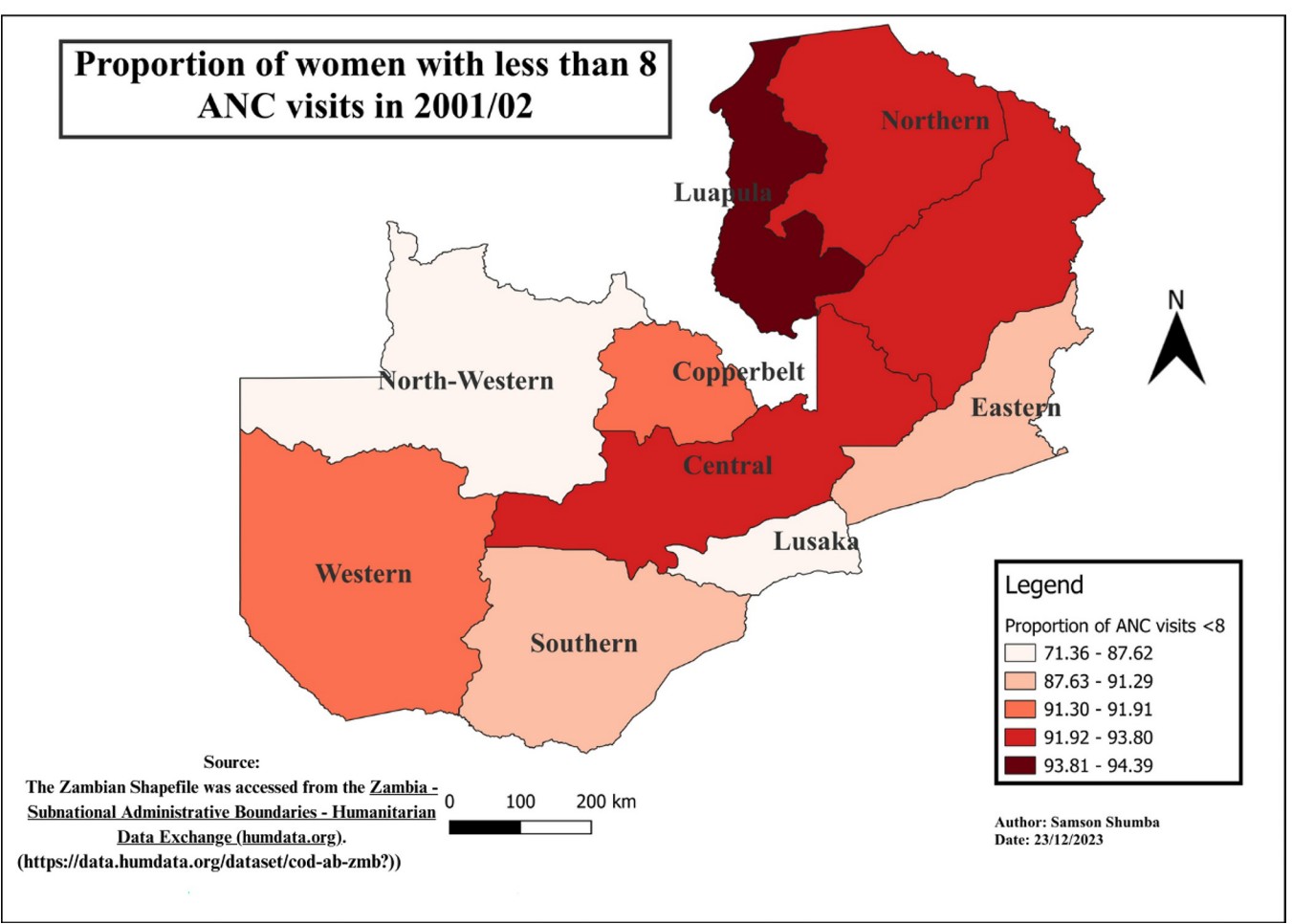

**Fig 6. Shows the spatial-temporal patterns of mothers with ANC visits less than 8 by provinces in Zambia in 2001/02.** The maps were generated using QGIS version 3.4.1. (Source: author generated).

The World Health Organization (WHO) initially advocated the Focused Antenatal Care (FANC) model in the 1990s, recommending a minimum of four ANC visits per pregnant woman [27]. However, in 2016, WHO revised its minimum recommended ANC visits from four to eight ANC visits. This change was reflected in the 2018 Demographic Health Survey. Given this transition, it's apparent that the 2016 WHO ANC model wasn't implemented before then, as the FANC model was in use. Consequently, the study is aware of the difference in required ANC visits between 2018 and preceding years [28, 29]. It is therefore, reasonable to assume that low attendance before 2018 adhered to the FANC model's minimum requirement of four visits, not eight. However, the study unveils a concerning trend: the proportion of ANC attendance fewer than eight visits has rather increased over time, indicating a decline in ANC attendance among pregnant women over the years. Proportion of delayed initiation of ANC visit in the study revealed an overall decline overtime this is similar to studies carried out in in Ethiopia [30–32] which showed that the trend for delayed ANC visits has been on the decline.

Furthermore, the study delved into the geospatial distribution of less than 8 ANC visits, revealing distinct patterns across provinces. The findings highlight that Western, Southern, Copperbelt, and Lusaka provinces consistently exhibited notably lower proportions of ANC visits less than 8 when compared to their counterparts. This gap could be attributed to varying

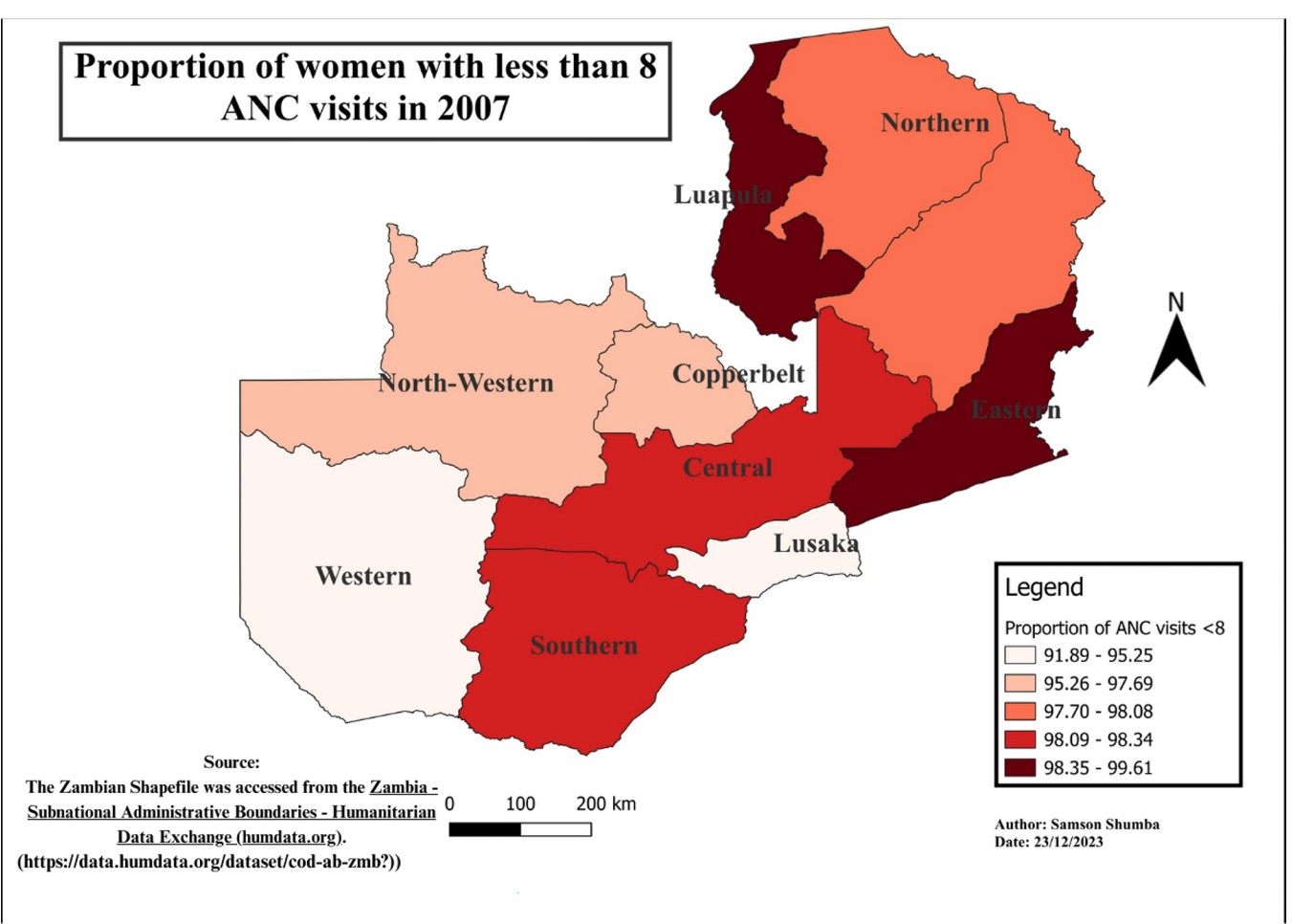

**Fig 7. Shows the spatial-temporal patterns of mothers with ANC visits less than 8 by provinces in Zambia in 2007.** The maps were generated using QGIS version 3.4.1. (Source: author generated).

levels of economic activities, employment rates, educational attainment, and the prevalence of urban landscapes over rural settings in these provinces [33]. Conversely, provinces such as Luapula, Muchinga, Northwestern, Northern, and Eastern reported considerably higher proportions of less than 8 ANC visits, pointing towards region-specific challenges that necessitate targeted interventions for improved maternal healthcare access and utilization.

Analyzing factors influencing inadequate ANC utilization (less than 8 ANC visits), the study identified education, wealth index, occupation, reading habits, television watching, and residing in the Copperbelt province as predictors of attending less than the recommended eight ANC sessions. These results align with those of previous studies, such as the work by Islam and Masud [25], indicating that these predictor variables were protective, correlating with a reduced risk of attending fewer than 8 ANC visits after accounting for other factors [34–36]. Notably, the exception was the frequency of watching television, which was associated with an increased risk. These results align with prior research, suggesting protective effects, with the exception of television watching, associated with increased risk. Additionally, stratified analysis across survey years indicates age and education as predictors, emphasizing the dynamic nature of ANC utilization over time. Consequently, Public health campaigns, leveraging media channels, should be intensified to educate and empower women, fostering health

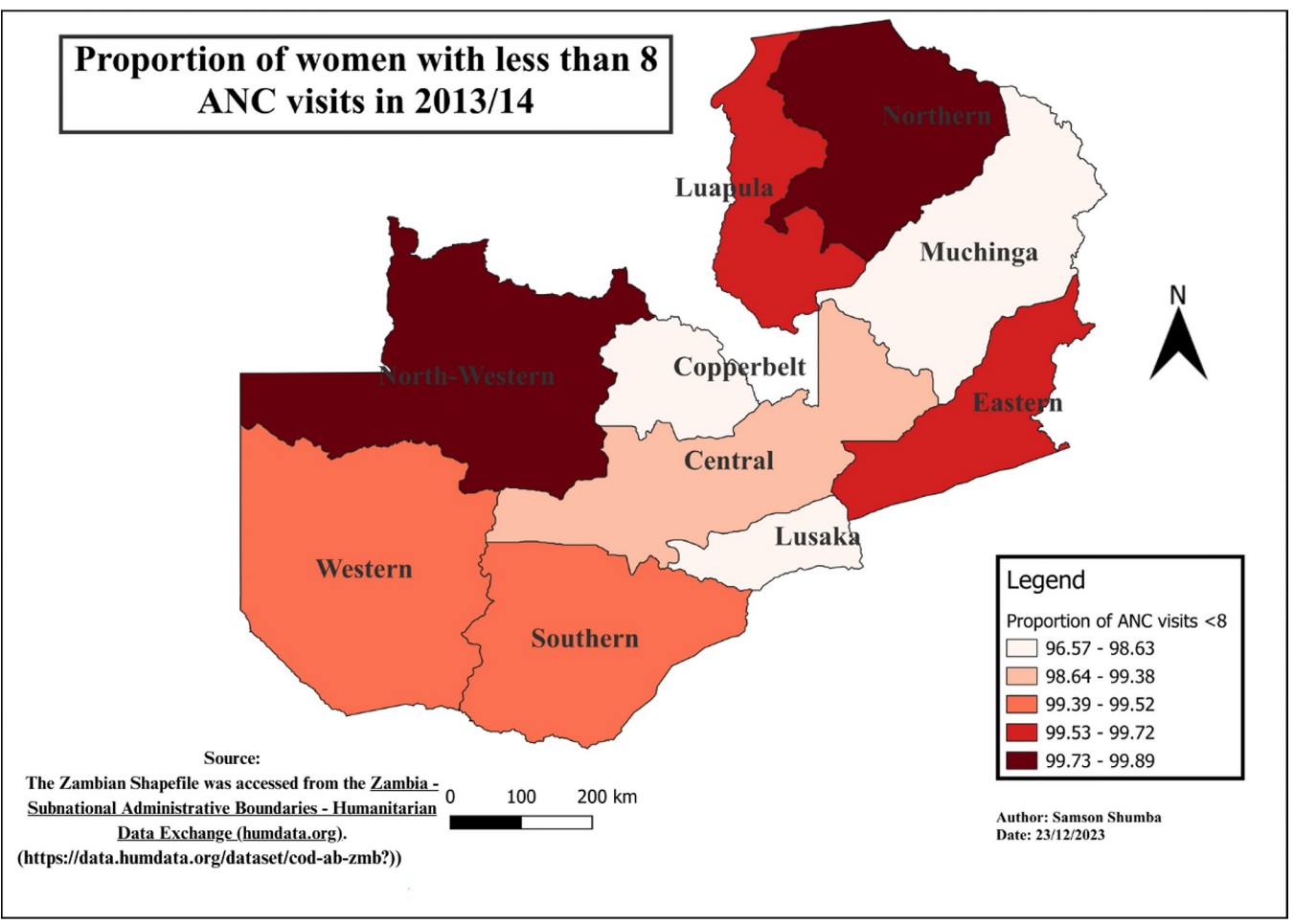

**Fig 8. Shows the spatial-temporal patterns of mothers with ANC visits less than 8 by provinces in Zambia in 2013/14.** The maps were generated using QGIS version 3.4.1. (Source: author generated).

literacy-related discussions and positively influencing their ANC behaviors. Additionally, healthcare providers should emphasize comprehensive ANC education, covering not only the quantity of visits but also the content and quality of services.

In contrast, a study in Rwanda revealed contradictory outcomes, indicating that employed women had reduced odds of attending all ANC sessions [37]. Conversely, our study aligns with research suggesting that regular reading of newspapers/magazines is associated with an increased likelihood of robust attendance at ANC visits. This association is attributed to higher literacy levels and education among women with access to newspapers, fostering health literacy-related discussions that positively influence their health-seeking behaviors [37–40]. Furthermore, a study in Papua New Guinea reported similar findings of employment and education level being protective of the risk of less than 8 ANC visits [40]. Additionally, a stratified analysis was conducted for each survey year from 1992 to 2018. The findings indicate that age predicted less than 8 ANC visits in 1992, 1996, and 2018. Education similarly served as a predictor, except in the year 2007. Intriguingly, the results reveal that education acted as a protective factor.

Furthermore, the study revealed majority of adolescents aged 15 to 19 (80.66%) delayed their first ANC visits compared to older women. Additionally, the study revealed that

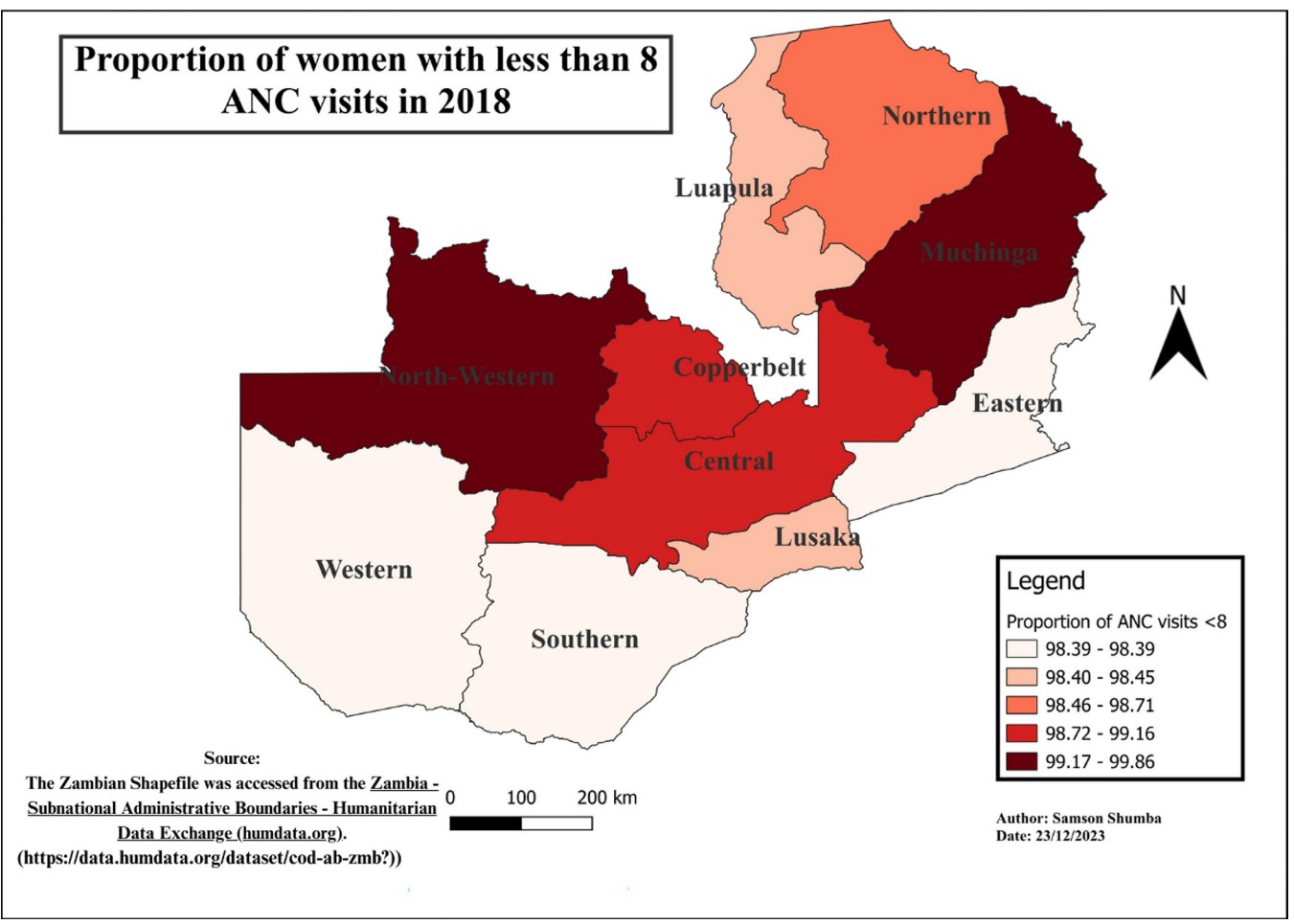

**Fig 9. Shows the spatial-temporal patterns of mothers with ANC visits less than 8 by provinces in Zambia in 2018.** The maps were generated using QGIS version 3.4.1. (Source: author generated).

**Table 5. Univariate and multivariable Generalized Linear Mixed Model of less than 8 ANC visit antenatal care utilization (combined analysis).**

| Variables | Univariate | | Multivariable | |
|---|---|---|---|---|
| | COR (95% CI) | P-value | AOR (95% CI) | P-value |
| **Age** | | | | |
| 15–19 | Ref (1) | | Ref (1) | |
| 20–24 | 0.83 (0.68–1.02) | 0.41 | 1.10 (0.66–1.86) | 0.72 |
| 25–29 | 0.71 (0.58–0.87) | 0.003 | 1.59 (0.92–2.74) | 0.10 |
| 30–34 | 0.58 (0.47–0.71) | <0.001 | 1.22 (0.70–2.11) | 0.49 |
| 35–39 | 0.62 (0.50–0.78) | 0.003 | 1.02 (0.58–1.80) | 0.95 |
| 40–44 | 0.64 (0.50–0.83) | 0.01 | 1.09 (0.56–2.11) | 0.80 |
| 45–49 | 0.61 (0.42–0.90) | 0.01 | 1.19 (0.43–3.32) | 0.74 |
| **Education level** | | | | |
| No Education | Ref (1) | | Ref (1) | |
| Primary | 0.84 (0.70–1.01) | 0.07 | 0.69 (0.39–1.22) | 0.21 |
| Secondary | 0.62 (0.51–0.76) | <0.001 | 0.78 (0.41–1.47) | 0.44 |
| Higher | 0.26 (0.19–0.35) | <0.001 | 0.30 (0.14–0.63) | 0.001* |

*(Continued)*

**Table 5.** (Continued)

| Variables | Univariate | | Multivariable | |
|---|---|---|---|---|
| | **COR (95% CI)** | **P-value** | **AOR (95% CI)** | **P-value** |
| **Marital status** | | | | |
| Not married | Ref (1) | | Ref (1) | |
| Married | 0.86 (0.76–0.98) | 0.02 | 0.76 (0.54–1.06) | 0.11 |
| **Wealth Index** | | | | |
| Poorest | Ref (1) | | Ref (1) | |
| Poorer | 0.91 (0.58–0.1.44) | 0.70 | 0.96 (0.60–1.53) | 0.86 |
| Middle | 0.91 (0.57–0.1.45) | 0.70 | 1.02 (0.63–1.67) | 0.94 |
| Richer | 0.57 (0.36–0.88) | 0.01 | 0.65 (0.36–1.15) | 0.14 |
| Richest | 0.25 (0.17–0.38) | <0.001 | 0.34 (0.17–0.70) | 0.003* |
| **Currently working** | | | | |
| No | Ref (1) | | Ref (1) | |
| **Yes** | 0.95 (0.86–1.04) | 0.27 | 0.66 (0.50–0.87) | 0.003* |
| **Frequency of listening to rad** | | | | |
| Not at all | Ref (1) | | Ref (1) | |
| <once a week | 0.89 (0.69–1.16) | 0.39 | 1.08 (0.68–1.72) | 0.75 |
| ≥ once a week | 0.93 (0.73–1.19) | 0.57 | 0.99 (0.67–1.46) | 0.95 |
| Almost everyday | 0.58 (0.48–0.71) | <0.001 | 0.78 (0.56–1.08) | 0.13 |
| **Frequency of reading NS/MZ** | | | | |
| Not at all | Ref (1) | | Ref (1) | |
| <once a week | 0.61 (0.48–0.77) | <0.001 | 0.77 (0.52–1.15) | 0.20 |
| ≥ once a week | 0.72 (0.55–0.95) | 0.02 | 1.02 (0.65–1.59) | 0.95 |
| Almost everyday | 0.36 (0.25–0.51) | <0.001 | 0.59 (0.35–0.97) | 0.04* |
| **Frequency of watching television** | | | | |
| Not at all | Ref (1) | | Ref (1) | |
| <once a week | 0.97 (0.67–1.40) | 0.86 | 1.95 (0.92–4.15) | 0.08 |
| ≥ once a week | 0.78 (0.56–1.08) | 0.14 | 1.18 (0.67–2.06) | 0.57 |
| Almost everyday | 0.63 (0.51–0.79) | <0.001 | 1.77 (1.11–2.80) | 0.02* |
| **Residence** | | | | |
| Urban | Ref (1) | | Ref (1) | |
| Rural | 3.20 (2.68–3.83) | <0.001 | 1.08 (0.73–1.61) | 0.68 |
| **Region** | | | | |
| Central | Ref (1) | | Ref (1) | |
| Copperbelt | 0.28 (0.19–0.39) | <0.001 | 0.47 (0.27–0.83) | 0.01* |
| Eastern | 1.40 (0.93–2.09) | 0.10 | 1.05 (0.56–1.98) | 0.87 |
| Luapula | 1.33 (0.88–2.03) | 0.18 | 1.80 (0.86–3.76) | 0.12 |
| Lusaka | 0.28 (0.19–0.40) | <0.001 | 0.73 (0.41–1.32) | 0.30 |
| Muchinga | 1.08 (0.72–1.63) | 0.70 | 0.80 (0.43–1.51) | 0.50 |
| Northern | 0.72 (0.48–1.07) | 0.10 | 0.91 (0.48–1.74) | 0.78 |
| Northwest | 0.89 (0.59–1.34) | 0.58 | 1.61 (0.79–3.28) | 0.19 |
| Southern | 1.15 (0.76–1.74) | 0.51 | 1.07 (0.57–2.01) | 0.84 |
| Western | 6.78 (3.12–14.72) | <0.001 | 1.29 (0.59–2.83) | 0.52 |

* = P-value <0.05 COR = Crude Odds Ratio AOR = Adjusted Odds Ratio

**Table 6. Multivariable Generalized Linear Mixed Model from 1992–2018 of frequency of antenatal care (Stratified analysis).**

| Variables | 1992 | 1996 | 2001/2 | 2007 | 2013/14 | 2018 |
|---|---|---|---|---|---|---|
| | AOR (95% CI) | AOR (95% CI) | AOR (95% CI) | AOR (95% CI) | AOR (95% CI) | AOR (95% CI) |
| **Age** | | | | | | |
| 15–19 | Ref (1) | Ref (1) | Ref (1) | Ref (1) | Ref (1) | Ref (1) |
| 20–24 | 0.68 (0.47–0.98)* | 0.85 (0.58–1.26) | 1.20 (0.84–1.72) | 0.83 (0.33–2.11) | 1.00 (0.32–3.14) | 1.42 (0.65–3.09) |
| 25–29 | 0.52 (0.36–0.75)* | 0.60 (0.41–0.89)* | 1.08 (0.75–1.56) | 0.95 (0.37–2.43) | 1.31 (0.41–4.23) | 2.94 (1.20–7.19)* |
| 30–34 | 0.44 (0.30–0.64)* | 0.54 (0.36–0.81)* | 0.70 (0.48–1.02) | 0.63 (0.24–1.64) | 0.80 (0.25–2.53) | 2.65 (1.05–6.65)* |
| 35–39 | 0.50 (0.33–0.75)* | 0.56 (0.36–0.87)* | 0.73 (0.48–1.10) | 0.64 (0.23–1.78) | 0.45 (0.14–2.99) | 2.63 (0.99–6.97) |
| 40–44 | 0.53 (0.32–0.86)* | 0.40 (0.24–0.66)* | 0.77 (0.47–1.27) | 0.71 (0.22–2.36) | 1.09 (0.11–10.76) | 1.45 (0.53–4.01) |
| 45–49 | 0.65 (0.29–1.44) | 0.39 (0.19–0.79)* | 0.57 (0.29–1.12) | 1.51 (0.17–13.20) | 1.09 (0.11–10.76) | 0.99 (0.25–3.99) |
| **Education level** | | | | | | |
| No Education | Ref (1) | Ref (1) | Ref (1) | Ref (1) | Ref (1) | Ref (1) |
| Primary | 0.78 (0.57–1.06) | 0.83 (0.60–1.17) | 0.80 (0.57–1.13) | 0.87 (0.40–1.91) | 0.66 (0.19–2.24) | 0.40 (0.12–1.33) |
| Secondary | 0.50 (0.35–0.72)* | 0.45 (0.31–0.65)* | 0.56 (0.38–0.84)* | 0.72 (0.29–1.80) | 0.49 (0.13–1.87) | 0.56 (0.15–2.05) |
| Higher | 0.39 (0.21–0.75)* | 0.16 (0.09–0.28)* | 0.30 (0.15–0.59)* | 0.32 (0.96–1.04) | 0.15 (0.03–0.67)* | 0.18 (0.04–0.83)* |
| **Marital status** | | | | | | |
| Not married | Ref (1) | Ref (1) | Ref (1) | Ref (1) | Ref (1) | Ref (1) |
| Married | 0.75 (0.59–0.95)* | 1.14 (0.91–1.43) | 1.00 (0.78–1.28) | 0.86 (0.50–1.47) | 0.63 (0.40–1.15) | 0.75 (0.42–1.35) |
| **Wealth Index** | | | | | | |
| Poorest | – | – | – | Ref (1) | Ref (1) | Ref (1) |
| Poorer | – | – | – | 0.71 (0.32–1.56) | 1.05 (0.40–2.75) | 1.05 (0.50–2.18) |
| Middle | – | – | – | 1.00 (0.41–2.42) | 0.72 (0.28–1.85) | 1.20 (0.52–2.79) |
| Richer | – | – | – | 0.55 (0.20–1.48) | 0.83 (0.26–2.64) | 1.01 (0.34–3.02) |
| Richest | – | – | – | 0.27 (0.08–0.90)* | 0.43 (0.11–1.73) | 0.63 (0.17–2.41) |
| **Currently working** | | | | | | |
| No | Ref (1) | Ref (1) | Ref (1) | Ref (1) | Ref (1) | Ref (1) |
| Yes | 0.97 (0.80–1.18) | 1.14 (0.91–1.43) | 1.23 (0.99–1.52) | 0.64 (0.41–1.01) | 0.68 (0.40–1.15) | 0.71 (0.44–1.15) |
| **Frequency of listening to rad** | | | | | | |
| Not at all | – | – | Ref (1) | Ref (1) | Ref (1) | Ref (1) |
| <once a week | – | – | 1.00 (0.73–1.38) | 1.13 (0.49–2.60) | 0.99 (0.45–2.19) | 1.44 (0.62–3.37) |
| ≥ once a week | – | – | 1.04 (0.75–1.44) | 0.59 (0.31–1.13) | 1.88 (0.87–4.04) | 1.16 (0.57–2.34) |
| Almost everyday | – | – | 0.67 (0.51–0.88) | 0.76 (0.43–1.32) | 1.21 (0.66–2.22) | 0.81 (0.43–1.52) |
| **Frequency of reading NS/MZ** | | | | | | |
| Not at all | – | – | Ref (1) | Ref (1) | Ref (1) | Ref (1) |
| <once a week | – | – | 0.72 (0.52–0.98) | 1.17 (0.59–2.32) | 0.47 (0.24–0.92)* | 1.26 (0.53–2.28) |
| ≥ once a week | – | – | 0.80 (0.53–1.19) | 1.72 (0.80–3.73) | 0.87 (0.39–1.93) | 0.78 (0.33–1.83) |
| Almost everyday | – | – | 0.36 (0.18–0.70) | 0.82 (0.35–1.94) | 0.45 (0.19–1.07) | 0.74 (0.23–2.42) |
| **Frequency of watching television** | | | | | | |
| Not at all | – | – | Ref (1) | Ref (1) | Ref (1) | Ref (1) |
| <once a week | – | – | 1.10 (0.73–1.38) | 1.98 (0.66–5.97) | 1.30 (0.36–4.45) | 3.85 (0.50–29.46) |
| ≥ once a week | – | – | 1.04 (0.75–1.44) | 1.14 (0.45–2.91) | 1.26 (0.44–3.65) | 0.73 (0.28–1.92) |
| Almost everyday | – | – | 0.67 (0.51–0.87)* | 1.64 (0.81–3.32) | 1.50 (0.63–3.57) | 0.97 (0.39–2.40) |
| **Residence** | | | | | | |
| Urban | Ref (1) | Ref (1) | Ref (1) | Ref (1) | Ref (1) | Ref (1) |
| Rural | 2.03 (1.44–2.87)* | 1.89 (1.43–2.30)* | 1.68 (1.21–2.34)* | 1.32 (0.64–2.72) | 0.77 (0.39–2.52) | 0.72 (0.34–1.50) |
| **Region** | | | | | | |
| Central | Ref (1) | Ref (1) | Ref (1) | Ref (1) | Ref (1) | Ref (1) |
| Copperbelt | 0.82 (0.50–1.34) | 0.44 (28–0.70)* | 0.64 (0.38–1.09) | 0.67 (0.28–1.60) | 0.31 (0.12–0.84)* | 0.69 (0.24–2.00) |

(*Continued*)

**Table 6.** (Continued)

| Variables | 1992 | 1996 | 2001/2 | 2007 | 2013/14 | 2018 |
|---|---|---|---|---|---|---|
| | AOR (95% CI) | AOR (95% CI) | AOR (95% CI) | AOR (95% CI) | AOR (95% CI) | AOR (95% CI) |
| Eastern | 0.93 (0.51–1.70)* | 1.49 (0.85–2.62) | 0.82 (0.47–1.42) | 1.37 (0.48–3.96) | 2.44 (0.58–10.28) | 0.55 (0.19–1.54) |
| Luapula | 0.70 (0.39–1.25)* | 0.82 (0.49–1.38) | 1.12 (0.59–2.13) | 4.22 (0.88–20.32) | 2.40 (0.57–10.17) | 0.97 (0.31–3.08) |
| Lusaka | 0.55 (0.33–0.92) | 0.53 (0.33–0.85)* | 0.33 (0.19–0.56)* | 0.78 (0.32–1.89) | 0.64 (0.22–1.84) | 0.90 (0.30–2.68) |
| Muchinga | – | – | – | – | 0.57 (0.19–1.73) | 1.01 (0.30–3.86) |
| Northern | 1.46 (0.77–2.76) | 0.84 (0.50–1.43) | 1.13 (0.66–1.95) | 0.94 (0.36–2.46) | 2.23 (0.52–9.56) | 0.71 (0.23–2.21) |
| Northwest | 0.61 (0.32–1.15)* | 0.54 (0.32–0.92)* | 0.25 (0.15–0.42)* | 0.81 (0.32–2.05) | 2.44 (0.58–10.34) | 6.20 (0.70–54.91) |
| Southern | 1.25 (0.71–2.21) | 0.49 (0.29–0.80)* | 0.71 (0.41–1.25) | 1.00 (0.39–2.57) | 1.09 (0.35–3.37) | 1.20 (0.48–6.97) |
| Western | 1.14 (0.60–2.19) | 0.62 (0.37–1.04) | 0.77 (0.42–1.41) | 0.71 (0.28–1.81) | 1.60 (0.42–6.15) | 0.80 (0.24–2.64) |
| ICC | 0.0841238 | 0.0394164 | 0.0755291 | <0.0001 | 0.1405596 | 0.2227786 |
| AIC | 3191.726 | 3126.52 | 3017.116 | 919.9457 | 860.0249 | 896.1335 |
| BIC | 3329.677 | 3267.639 | 3215.038 | 1134.642 | 1116.486 | 1144.401 |

\* = P-value <0.05

education, residence (rural and urban) and region were associated with delayed initiation of first ANC visit. These findings are similar to studies done in various countries that revealed that majority of women initiate first antenatal visit very late after the first trimester, which is against the WHO recommendation [41–43]. A study done in Wollaita Soddo town revealed that education was associated with timing of first ANC visit. Women with secondary and higher education were more times likely to initiate than those with primary and who didn't have formal education [44], similar with a study conducted in Addis Ababa [45].

The stratified analysis revealed that only education consistently predicted the underutilization of ANC visits (less than 8 visits) from 1992 to 2018, except for 2007, where it showed no significant effect. Remarkably, education acted as a protective factor throughout these years. Previous studies have similarly highlighted the significant role of education in determining optimal ANC utilization [46–49]. This association may stem from educated women's likely possession of adequate knowledge regarding ANC services and their understanding of the importance of early booking and attending the recommended ANC visits. Moreover, education may empower some women to overcome gender-specific discrimination and barriers, such as domestic challenges [50, 51].

## Knowledge gaps

This study reveals several critical knowledge gaps regarding antenatal care (ANC) utilization in Zambia. Notably, there is inadequate implementation of WHO recommendations, as evidenced by an increasing proportion of women reporting fewer than the recommended eight ANC visits, indicating insufficient adherence to guidelines aimed at enhancing maternal healthcare. Additionally, persistent urban-rural disparities in ANC utilization highlights a lack of targeted interventions to improve access and quality of care in rural areas, emphasizing the need for tailored healthcare strategies that address geographical and socio-economic barriers. The study also uncovers a counterintuitive finding regarding media consumption; specifically, television watching is associated with an increased risk of attending fewer than eight ANC visits, suggesting a need for further exploration of how media affects health-seeking behaviors. While education consistently serves as a protective factor in ANC utilization, the research indicates that it does not fully capture the complexity of influences on ANC attendance, warranting further investigation into how education interacts with other socio-cultural factors. Lastly,

despite a decline in delayed initiation of ANC visits, a significant number of adolescents still start ANC late, underscoring a gap in awareness and education regarding the importance of timely ANC initiation and the need for more effective public health campaigns targeting young women.

## Limitations and strengths of the study

This study draws strength from the utilization of national data, providing a representative sample of the adolescent female population aged 15 to 49 in Zambia. Consequently, the study's findings are applicable and can be generalized to the specified target population of women with prior childbirth within this age range. However, it is essential to acknowledge the study's limitations. The reliance on the latest Zambia Demographic and Health Survey (ZDHS) dataset from 2018 follows a cross-sectional study design, implying that the results indicate correlation rather than causation between the outcome of interest and individual or contextual factors. Additionally, caution is advised when extending the findings to the broader adolescent age group of 10 to 19 years. Moreover, the contextual factors utilized in the study are derived from the ZDHS, potentially limiting their ability to fully capture the community experience. These considerations are crucial for a nuanced interpretation of the study's outcomes.

## Conclusion

In summary, this study sheds light on the critical challenges surrounding antenatal care (ANC) utilization in Zambia, exposing a concerning upward trend in less than 8 ANC visits, especially in meeting the prescribed eight ANC visits. The results underscore a substantial disparity between observed ANC practices and the established guidelines set by the World Health Organization Moreover, it notes a decreasing trend in the delay of initiating the first ANC visit from 1996 to 2018, even though it is still worrying. Persistent urban-rural discrepancies highlight the urgent need for tailored interventions aimed at rectifying significant deficiencies in ANC access and utilization across diverse settings. Notably, the study brings to light a noteworthy improvement in specific healthcare aspects, exemplified by a significant rise in HIV testing rates within ANC services. However, the observed decline in discussions about HIV transmission and preventive measures emphasizes the necessity of adopting a comprehensive approach to ANC, addressing both the quantity and quality of services. Additionally, the study identifies education, wealth index, occupation, reading habits, television watching, and residing in the Copperbelt as predictive factors for attending fewer than the recommended eight ANC sessions.

## Supporting information

**S1 Table. Region of women who have previously given birth, linked with delayed initiation of their first antenatal care (ANC) visit.**
(DOCX)

## Author Contributions

**Conceptualization:** Samson Shumba.

**Data curation:** Samson Shumba.

**Formal analysis:** Samson Shumba.

**Methodology:** Samson Shumba, Isaac Fwemba, Violet Kaymba.

**Software:** Samson Shumba.

**Supervision:** Isaac Fwemba, Violet Kaymba.

**Validation:** Isaac Fwemba, Violet Kaymba.

**Visualization:** Isaac Fwemba, Violet Kaymba.

**Writing – original draft:** Samson Shumba.

**Writing – review & editing:** Isaac Fwemba, Violet Kaymba.

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
