## [Decision Letter · Decision Letter 0]

22 Jul 2024

PGPH-D-24-00899

Analyzing Predictors and Geospatial Trends of the Number of Visits and Timing of Antenatal Care in Zambia: A Generalized Linear Mixed Model (GLMM) Investigation from 1992 to 2018

Dear Dr. Shumba,

Thank you for submitting your manuscript to PLOS Global Public Health. After careful consideration, we feel that it has merit but does not fully meet PLOS Global Public Health’s publication criteria as it currently stands. Therefore, we invite you to submit a revised version of the manuscript that addresses the points raised during the review process.

The manuscript has been reviewed by two reviewers and their comments are provided below. They have a number of recommendations to improve the quality of the study. In particular, they request revisions to further justify the analytical approach, refine the analysis, and improve the quality of the reporting. Could you please carefully address the concerns raised?

We look forward to receiving your revised manuscript.

Kind regards,

Marianne Clemence

Staff Editor

Journal Requirements:

Additional Editor Comments (if provided):

Reviewers' comments:

Reviewer's Responses to Questions

**Comments to the Author**

1. Does this manuscript meet PLOS Global Public Health’s publication criteria? Is the manuscript technically sound, and do the data support the conclusions? The manuscript must describe methodologically and ethically rigorous research with conclusions that are appropriately drawn based on the data presented.

Reviewer #1: Yes

Reviewer #2: Yes

2. Has the statistical analysis been performed appropriately and rigorously?

Reviewer #1: Yes

Reviewer #2: No

3. Have the authors made all data underlying the findings in their manuscript fully available (please refer to the Data Availability Statement at the start of the manuscript PDF file)?

Reviewer #1: Yes

Reviewer #2: Yes

4. Is the manuscript presented in an intelligible fashion and written in standard English?

Reviewer #1: Yes

Reviewer #2: Yes

5. Review Comments to the Author

Reviewer #1: Thank you for the opportunity to review this paper. The study aims to analyze the predictors and geospatial trends of the number of visits and timing of antenatal care in Zambia from 1992 to 2018, utilizing a generalized linear mixed model (GLMM) approach. The paper is well-written; however, the clarity and transparency of the methods and presentation of results can be substantially improved.

Major comments

1. Explain the rationale for choosing the GLMM model in your analysis. Describe why this model is appropriate for your data and research objectives

2. Provide a clearer explanation of the model. Include the model equation, specifying which variables or levels were considered as random effects in your GLMM model. This will aid in understanding your analytical approach.

3. Clarify whether enumeration areas (EAs) were considered as random effects in your model. If survey weights were used in the regression, mention this explicitly. If not, provide a justification for their exclusion.

4. Given the extensive study period from 1992 to 2018, address potential variations by including the survey period as a random effect in your model. This adjustment can account for changes over time instead of considering the study period as a predictor variable.

5. In Table 3, explain why data from 1992 to 2001-2 were excluded from the multivariate model, and only data from 2007 to 2018 were used. Provide a rationale for this decision.

6. In the results section under "Quality of antenatal care offered between 2007 and 2018," you mention a significant increase in HIV testing rates from 46% to 98%. Specify the statistical test used to determine this significance and provide the corresponding p-value.

Minor comments

1. Improve the quality of figures (Figure 1, Figure 2, Figure 3, and Figure 4) by uploading high-resolution images. Enhance the readability of legends, particularly in Figure 4.

2. Add details of each survey date and the number of women interviewed in Figure 1. Since different study period survey data were pooled, providing all this information in a single figure will make it more self-explanatory.

3. Consider using a line chart instead of a bar chart to visually represent trends over time more effectively. Experiment with this change to see if it improves visual clarity.

4. Ensure that tables are ordered sequentially. There is a duplication of Table 3; rectify this by renaming or renumbering as necessary.

5. Make the Table 3 (second one) self-explanatory. Ensure proper alignment of categories so that each variable is distinguishable from others.

6. In Table 2, Table 3, and Supplementary Table 1, ensure that individual "n" values add up to the total "n" for all years. Currently, the sample sizes do not match when pooling all survey data.

7. Clarify why the study extracted pertinent variables only from the women's data files (individual recode) in the 2018 ZDHS dataset. Explain why data from other study years were not included.

Reviewer #2: Comments on

Title

1. The title is vague (not self -explanatory): Better to re-write in a way that clearly indicates the outcome of interest (inadequate ANC utilization)

2. The paper analyses temporal trend and special distribution of number of ANC visits but the title doesn’t indicate that

Abstract

1. The aim of the study is a mix-up of number of visit, timing of ANC, predictors, geospatial analysis, utilization, needs revision.

Background

1. The first sentence is far from the aim of the study. Better to remove

2. Line 7-11 of 1st paragraph confuses readers by merging causes of maternal and new-born mortality. Better to split the sentence

3. 2nd paragraph-n sentences need bracket and full stops.

4. Line 8 of 3rd paragraph doesn’t indicate which initiation rate: timely, global?

5. Lastly, what was the knowledge gap which justifies the originality/conduct of this study?

Methods

1. Data analysis section:

a. It would have been good if the model will Poisson for the number of ANC visits (count data)

b. Or the category should align with the policy recommendations as before 2016 (or the country adopting time) and after to say ANC is underutilized.

c. Is it justifiable to use chi-square test over regression model

d. Better to come up with proper model fitness tests for categorized GLMM model.

e. On model selection section: Better to remove statistical jargon table 2 (diagnostic tests) for non-statistician readers and indicate your ICC value to see the warrantee for multilevel regression model.

Result

1. Quality of ANC … In my opinion those are the components of ANC visits which are believed to be increased when number of visits improved. These my not reflect the “quality of care” so, better to state as ANC contents

Discussion

1. Paragraph 6, reveals there is special variation of ANC visit and speculated that the variation might be due to factors as economic, employment, etc.

a. Is it not possible to include those factors for the predictor analysis?

b. Could you recommend further study to investigate the effects of those variables

c. Did you consider geographically weighted regression method for such ecological data?

6. PLOS authors have the option to publish the peer review history of their article (what does this mean?). If published, this will include your full peer review and any attached files.

**Do you want your identity to be public for this peer review?** For information about this choice, including consent withdrawal, please see our Privacy Policy.

Reviewer #1: No

Reviewer #2: **Yes: **Dr Asressie Molla

---

## [Decision Letter · Decision Letter 1]

11 Sep 2024

Spatial-Temporal Patterns and Predictors of Timing and Inadequate Antenatal Care Utilization in Zambia: A Generalized Linear Mixed Model (GLMM) Investigation from 1992 to 2018

PGPH-D-24-00899R1

Dear Mr. Shumba,

We are pleased to inform you that your manuscript 'Spatial-Temporal Patterns and Predictors of Timing and Inadequate Antenatal Care Utilization in Zambia: A Generalized Linear Mixed Model (GLMM) Investigation from 1992 to 2018' has been provisionally accepted for publication in PLOS Global Public Health.

Best regards,

Julia Robinson

Executive Editor

Reviewer Comments (if any, and for reference):

Reviewer's Responses to Questions

**Comments to the Author**

1. If the authors have adequately addressed your comments raised in a previous round of review and you feel that this manuscript is now acceptable for publication, you may indicate that here to bypass the “Comments to the Author” section, enter your conflict of interest statement in the “Confidential to Editor” section, and submit your "Accept" recommendation.

Reviewer #1: All comments have been addressed

2. Does this manuscript meet PLOS Global Public Health’s publication criteria? Is the manuscript technically sound, and do the data support the conclusions? The manuscript must describe methodologically and ethically rigorous research with conclusions that are appropriately drawn based on the data presented.

Reviewer #1: Yes

3. Has the statistical analysis been performed appropriately and rigorously?

Reviewer #1: Yes

4. Have the authors made all data underlying the findings in their manuscript fully available (please refer to the Data Availability Statement at the start of the manuscript PDF file)?

Reviewer #1: Yes

5. Is the manuscript presented in an intelligible fashion and written in standard English?

Reviewer #1: Yes

6. Review Comments to the Author

Reviewer #1: (No Response)

7. PLOS authors have the option to publish the peer review history of their article (what does this mean?). If published, this will include your full peer review and any attached files.

**Do you want your identity to be public for this peer review?** For information about this choice, including consent withdrawal, please see our Privacy Policy.

Reviewer #1: No
